# Egg masses as training aids for spotted lanternfly *Lycorma delicatula* detection dogs

**Jennifer L. Essler**[1]*, **Sarah A. Kane**[1], **Amanda Collins**[1], **Kaley Ryder**[1], **Annemarie DeAngelo**[1], **Patricia Kaynaroglu**[1], **Cynthia M. Otto**[1,2]

1 Penn Vet Working Dog Center, School of Veterinary Medicine, University of Pennsylvania, Philadelphia, Pennsylvania, United States of America, 2 Department of Clinical Science and Advanced Medicine, School of Veterinary Medicine, University of Pennsylvania, Philadelphia, Pennsylvania, United States of America

* esslerj@vet.upenn.edu

**Data Availability Statement:** All relevant data are within the manuscript and its Supporting Information files.

## Abstract

The spotted lanternfly (*Lycorma delicatula*) is an invasive species first detected in 2014. The insect feeds on plants causing severe damage in vineyards such as the occurrence of sooty mold fungus that impairs leaf photosynthesis. Currently, there is extensive research on how to track and ultimately prevent the spread of this species. It lays eggs that persist through the winter, while the adults die out, which presents a unique opportunity to enter infested or suspected infested areas to begin quarantine and management of the spread while the species is dormant. Detection dogs may be a tool that can be used to search out the spotted lanternfly egg masses during this overwintering period, however it is not known whether dogs can detect any specific odor from the spotted lanternfly eggs. Moreover, as the eggs are only available during certain times of the year, and hatch based on temperature, finding training aids for the dogs could prove difficult. In this study, we investigated whether three detection dogs could learn the odor from dead spotted lanternfly egg masses and if so, whether that would allow them to recognize live spotted lanternfly egg masses. We found that dogs could be trained to find dead spotted lanternfly egg masses, and could learn to ignore relevant controls, with high levels of sensitivity and specificity (up to 94.6% and 92.8%, respectively). Further, we found that after the training, dogs could find live spotted lanternfly egg masses without additional training and returned to previous levels of sensitivity and specificity within a few sessions. Coded videos of training and testing sessions showed that dogs spent more time at the egg masses than at controls, as expected from training. These results suggest that dead spotted lanternfly egg masses could be a useful training aid for spotted lanternfly detection dogs.

## Introduction

The spotted lanternfly *Lycorma delicatula* (White) (Hemiptera: Fulgoridae) [1] is a planthopper insect native to China. It has since become an invasive species in the United States, first reported in 2014 in Pennsylvania [2, 3]. The spotted lanternfly feeds on both herbaceous and woody plants by eating the sap. This feeding can weaken and stress the plants, and further

**Funding:** CMO received USDA (United States Department of Agriculture) grant AP19PPQS&T00C206 which funded this project. The funders had no role in study design, data collection and analysis, decision to publish, or preparation of the manuscript.

**Competing interests:** The authors have declared that no competing interests exist.

damage is induced by the sooty mold development that occurs after the plant gets covered with the honeydew, or sugary excrement, excreted by the spotted lanternfly, which can block photosynthesis on understory plants [4]. The spotted lanternfly feeds on a wide range of hosts, though they appear to have a strong preference for the tree of heaven, *Ailanthus altissima* (Sapindales: Simaroubaceae) [4]. Thus far, the estimated expected overall annual direct economic impact of spotted lanternfly damage on Pennsylvania agriculture is at least $42.6 million statewide. The most heavily impacted agricultural operations are nurseries, fruit growers, particularly grapes, and Christmas tree growers [5].

The spotted lanternfly lays eggs in one season per year, typically from September until early November, but has been seen as late as December in Pennsylvania. Egg masses contain 30 to 50 eggs, are covered in a yellow-brown waxy covering [3], and vary in size but are typically about one inch in length. Eggs are typically laid on trees, up to 17 meters above the ground on the tree of heaven, their most preferred host plant [6]. However, egg masses can be readily found on other substrates [7], including natural and unnatural items such as stone, automobiles, rail cars, and shipping pallets [4], all quite low to the ground (Fig 1A and 1B). Multiple egg masses can be found on each tree or substrate. These eggs overwinter, waiting out the winter season, and the nymphs are seen in May [3, 8]. There are many directions in which researchers and technical professionals in the field are taking to investigate potential ways to reduce the threat and spread of the spotted lanternfly [7, 9, 10], and this 5 to 8 month period in which the adults die out and the eggs overwinter is when the spread of these egg masses could most easily go undetected. Thus, implementation of tools which can prevent the spread of this species, in particular when they are in the egg stage, could be a promising pest management approach, reducing its serious damage on host plants.

Detection dogs may be used to search out the spotted lanternfly egg masses during this overwintering period by screening cargo, rail and automobiles, among other items, for evidence of egg masses in order to limit transport of this invasive species. These transportation means have already been highlighted for their potential role in the spread of the spotted lanternfly [11], while railways in particular have been indicated as vectors for invasive species spreading in other species [12, 13]. Canines are used in many domains to detect different odors, including drugs [14], land mines [15], accelerants [16], hazardous materials [17], human scent trails [18], human remains [19], live animal scents [20, 21], and animal scat [22], among other things (see [20] for a review). Dogs have been shown to be useful tools in the detection of other insect egg masses, as Wallner and Ellis [23] showed that dogs could be trained to detect both the pheromones and the egg masses of the gypsy moth, *Lymantria dispar L.* (Lepidoptera: Erebidae). In that study, dogs were trained to find the gypsy moth egg masses initially in the laboratory, but eventually in a field setting, with success. Gottwald and colleagues [24] report that dogs can detect infections of citrus huanglongbing (citrus greening disease) long before the disease can be seen either visually or with alternative molecular methods. Lee and colleagues [25] were able to train dogs to find the invasive brown marmorated stink bug in the laboratory, as well as at field sites, during its overwintering. Moreover, the dogs are able to search areas with relative speed, compared to humans. Finally, though a variety of insecticides have been shown to be effective against the spotted lanternfly [26], this is not a solution for potentially infected areas such as buildings (inside) and many types of cargo, leaving a multitude of scenarios where detection dogs may be particularly helpful.

The most obvious hurdle in training dogs on spotted lanternfly egg masses is the lack of a training tool to do so. First, the egg masses are only found during part of the year before hatching in May [3], and second, housing of live egg masses likely incurs significant levels of biosecurity to ensure that the eggs are neither transported and released into new areas, nor that they reach temperature levels high enough to cause hatching outside of the quarantine zone [27].

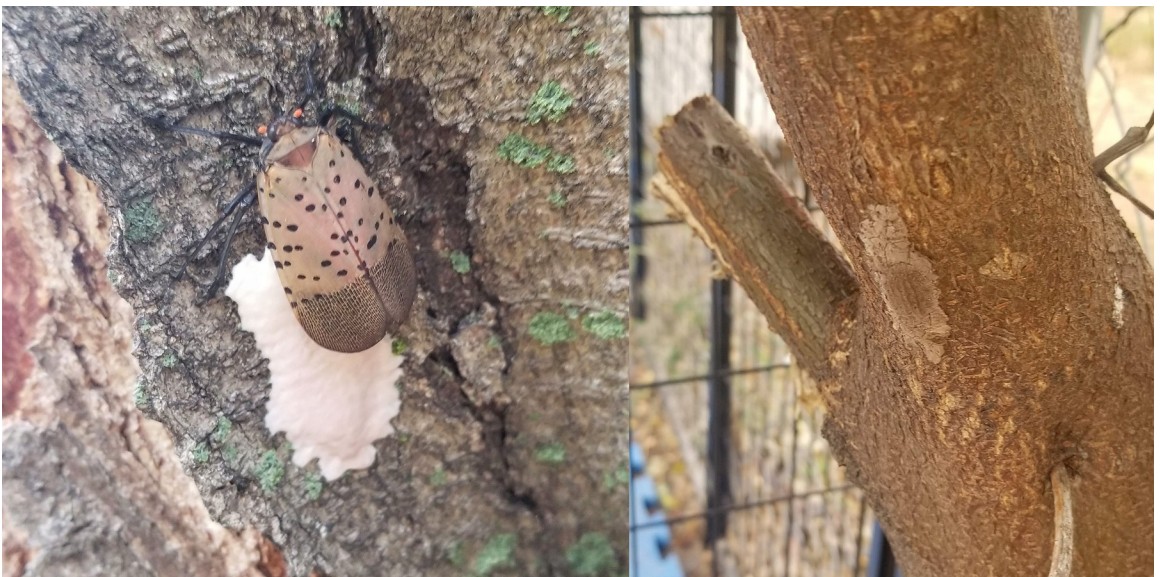

**Fig 1.** (A) Spotted lanternfly laying fresh eggs on tree bark. (B) Spotted lanternfly eggs with mud-like covering.

One potential solution to this problem would be to train on dead egg masses, if the odor were similar enough that the dogs could transfer to live egg masses with relative ease.

In this study, we aimed to test whether detection dogs trained to detect dead spotted lanternfly egg masses could transfer this training to detecting live egg masses, spontaneously or with relative ease. Our first aim was to train the dogs to detect dead egg masses attached to bark as well as dead egg masses separated from the bark, to ensure that the dogs were using the egg mass odor to distinguish barks with masses attached from barks that were egg mass-free. If this training proved successful, we aimed to test whether the dogs would spontaneously detect live egg masses, both attached to bark as well as separated from the bark.

## Materials & methods

### Ethics statement

This study was approved by the University of Pennsylvania Institutional Care and Use Committee for university-owned canines (Protocol #806730) and privately-owned canines (Protocol #806741). All dogs lived with either a foster family or their owner and were brought into the center for training. Privately owned dogs' owners signed an informed-consent form. All dogs' participation for every session was considered voluntary, as they were allowed to refuse to participate at any time.

### Subjects

Three trained detection dogs were used in this study. One dog was in training at the Penn Vet Working Dog Center, one dog was a graduate of the working dog center, and one was a privately-owned detection dog (Table 1). All dogs had previously been used in other odor detection studies at the center, or in their career. The previously trained odors as well as their trained final alert behaviors for each dog are listed. Training occurred up to three days (or three sessions) each week.

**Table 1. Detection dogs used in this study.**

| Dog | Sex | Age | Breed | Trained Odors | Final Response | Status |
|---|---|---|---|---|---|---|
| **Toby** | M | 1.5 years | Small Münsterländer | UDC* | Stand-Stare | PVWDC Dog |
| **Grizzly** | M | 5 years | German Shepherd | Pottery | Stand-Stare | Private Dog |
| **Pacy** | F | 6 years | Labrador Retriever | Human Remains, Pottery | Sit | Private Dog |

* All dogs at the Penn Vet Working Dog Center (PVWDC) are trained to search for odor using a training compound called Universal Detection compound (UDC) [28, 29].

## Experimental setup

The target odor was placed in an 8-port stainless-steel scent wheel approximately 2.1 meters in diameter (Fig 2). Each of the eight arms held one port, and an attached steel plate to limit the scent cone overlap between ports. A random number generator was used to determine which port would hold the target odor for each trial, with all ports equally likely to receive the target sample once during each trial. However, it was never in the same port more than two trials in a row. Vinyl gloves were used when handling samples and touching the wheel, and gloves were changed between handling samples. The tops of the ports, where the dogs' noses could touch, were wiped between trials with a paper towel to remove any excess saliva left while searching or performing their trained final response. The whole wheel was cleaned between dogs, and the room in which the wheel was located was cleaned daily, with 70% isopropyl alcohol.

The wheel was located behind a barrier and monitored with a video feed so that all humans were out of sight of the dogs, and the dogs could not use any visual cues from the humans. Moreover, a white noise machine was played to help ensure that dogs could not use any auditory cues from the humans. Dogs were trained to search the scent wheel ports in order, and were not allowed to return back to odors (e.g. if a port was passed, they could not return to it), and all three dogs had previously been trained to identify a wheel without any target sample

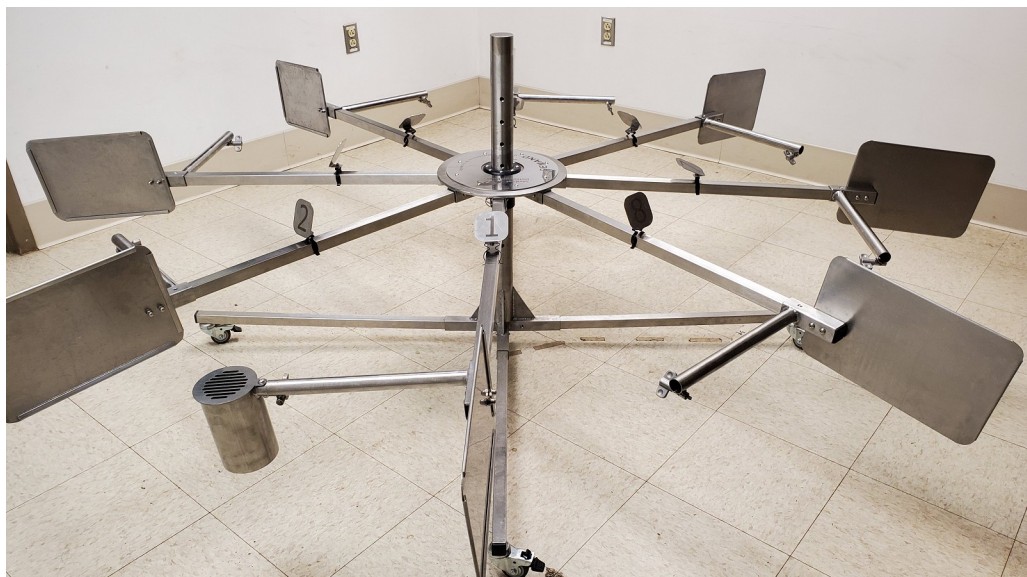

**Fig 2. Photograph of the 8-port stainless-steel scent wheel used in this study (manufactured by DEMAND (design & manufacture for disability), Abbots Langley, UK, http://www.demand.org.uk).** Ports which held the odors moved between arms of the wheel for example, in "1" there is a port present–all other arms are empty.

present (a blank wheel) by leaving the room, thus avoiding a forced-choice paradigm. Video records were taken for all trials, along with written records of the dogs' behavior at each port. Three behaviors were noted on target odors: alert (the dog showed a full trained final response), hesitate (the dog spent more time at an odor but did not show a full trained final response), and pass (the dog passed the odor with no distinguishable change of behavior). If a dog showed a trained final response on a non-target odor, this was marked as a false alert. If the dogs hesitated at any odor, or gave a false positive, the behavior was ignored. If the dogs gave a trained final response at a target odor, the trainer could see it via the video feed, and would mark the behavior with an auditory marker (a clicker) and then reward the dog when it returned from the wheel (Fig 3). Alert was defined differently for each dog and depended on their prior training as well as final trained response behavior. Pacy with a sit behavior had to sit in front of the target port in order for it to be categorized as an alert. Grizzly and Toby each had a stand-stare response and were required to stand with their nose at port for average of three and four seconds, respectively, in order for it to be categorized as an alert (a stopwatch was utilized; trial by trial the alert length varied in order to ensure the dog was not waiting for a specific number of seconds). Hesitates referred to a change of behavior at the port that was distinguishable compared to the dog's normal searching behavior, involving increased time spent at the port sniffing. Sniffing behavior has previously been shown as a relevant behavior to the odor type in detection dogs [30]. The experimenter, who set out the odors for each trial, was not blind to the placement of the odors.

## Training target odor–dead egg masses

Egg masses scraped off bark (Fig 4A), as well as egg masses attached to bark (Fig 4B), were frozen in a -80 degree Celsius freezer for 96 hours to ensure that the eggs were dead. This was a duration that entomologists at the USDA felt confident would devitalize the eggs (Gregory R. Parra, USDA, Personal Communication). Training samples were kept in 16-ounce mason jars in a refrigerator maintained between 0 and 4 degrees Celsius when not being used for training. Dogs were trained on seven different egg masses attached to bark and one egg mass scraped

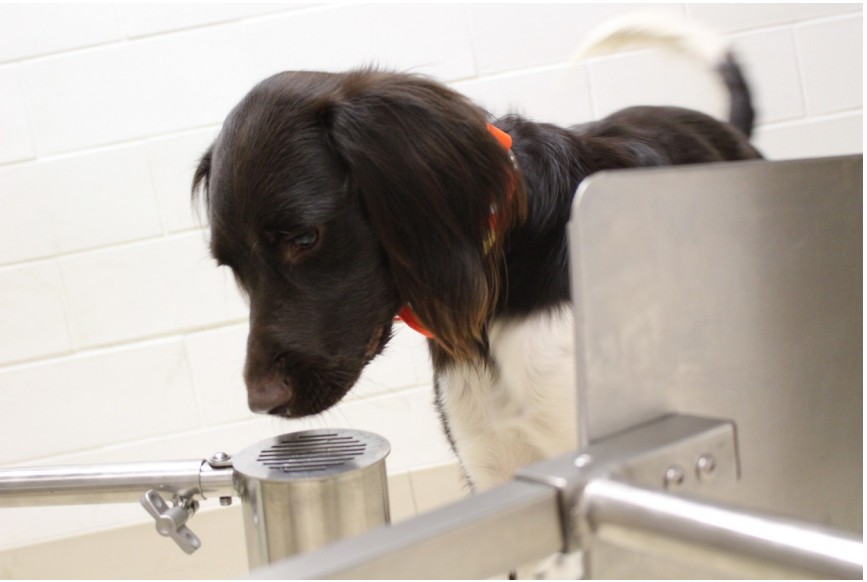

**Fig 3. Detection dog Toby giving a final response (stand-stare) at target odor.**

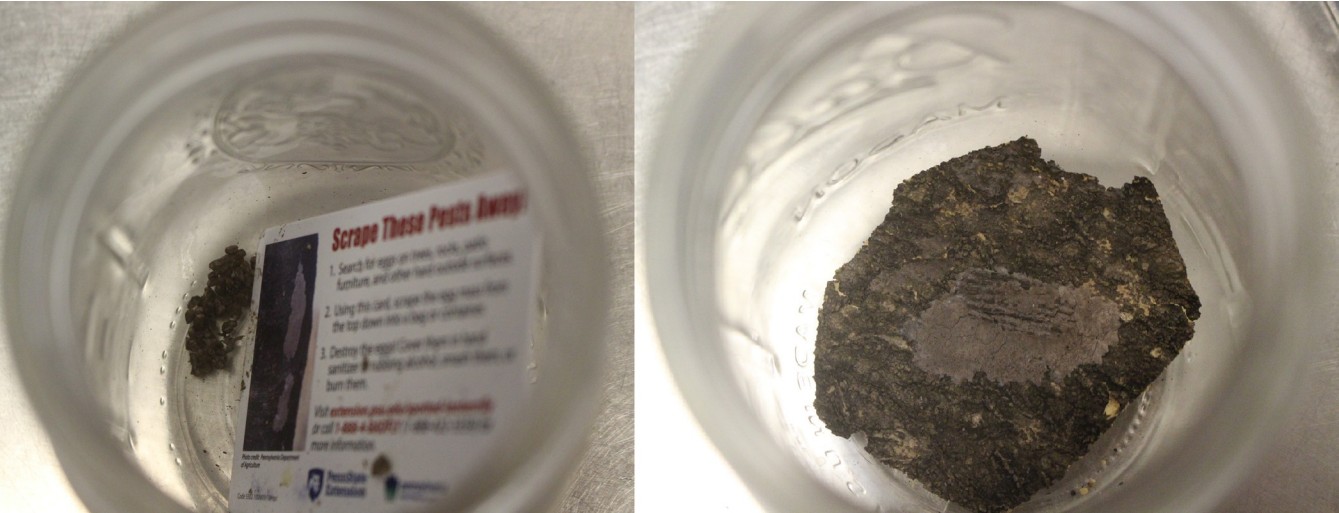

**Fig 4.** Pictures of the training target odor of dead spotted lanternfly egg masses–scraped eggs off bark (A) and eggs on bark (B).

from the bark. The egg masses attached to bark were from the *A. altissima* tree, while the egg mass scraped from bark was scraped from the *Prunus serrulata* (Rosales: Rosaceae), or Japanese flowering cherry.

## Control odors

In order to ensure that the dogs were recognizing the odor of the egg masses specifically, controls were introduced immediately. The main control for this study was egg mass-free bark from the *A. altissima* tree (Fig 5), but we introduced barks from multiple other trees and plants as well, including: *Fraxinus americana*, mulberry tree from the genus *Morus* (Rosales: Moraceae), honeysuckle plant from the genus *Lonicera* (Dipsacales: Caprifoliaceae), cherry blossom

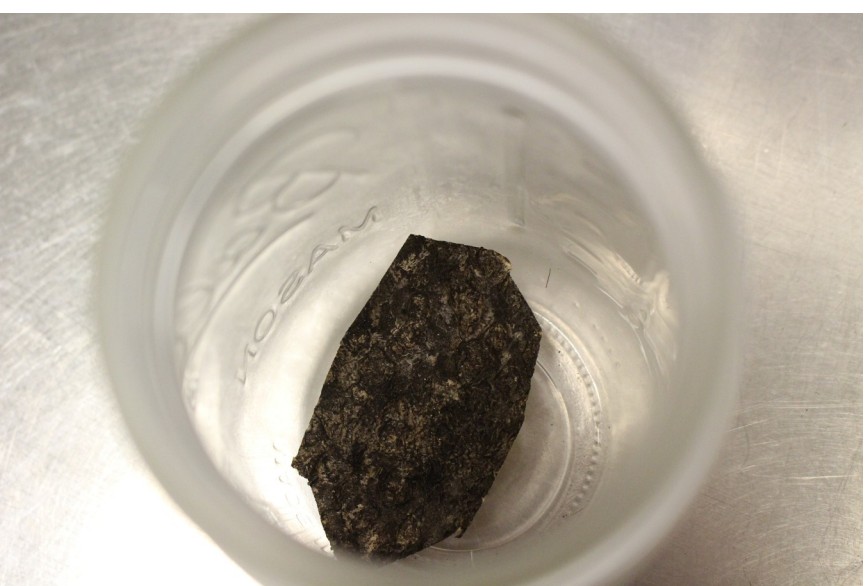

**Fig 5. Picture of one of the control barks from *Ailanthus altissima*.**

trees from the genus *Prunus*, and *Elaeagnus umbellata* (Rosales: Elaeagnaceae). These were all trees or plants that the spotted lanternfly egg masses were seen on during sample collection, many of which have already been reported as host plants for this species [3, 31].

The number of bark controls placed into the wheel increased as the dogs became proficient at recognizing their target odor, until six of the other arms (less the one containing the dead egg mass, and one other control, see below) were bark controls. Barks from other plant species were introduced at various stages of the training, to ensure that the dogs continued to ignore mass-free bark generally, and only one new bark was introduced within each training session. As we were unsure whether the freezing process changed the odor of the egg masses as well as the bark, bark controls were presented that had gone through the freezing process as well as those that had not.

Other than the egg mass-free bark as a control, multiple other introduced controls included items that may have come into contact with the egg mass while they were being collected. These controls included plastic scrapers, gloves (vinyl and nitrile), plastic shipping material, paper towels, cardboard, tape, and plastic storage containers (Fig 6). These controls were clean and free of spotted lanternfly odor. Finally, after training on these controls, an egg mass from a separate species (European Mantis, *Mantis religiosa (L.)* (Mantodea: Mantidae)) was added to ensure the dogs were not alerting to insect eggs generally. These controls were presented multiple months after the last training session of the dogs, due to the COVID-19 pandemic, to determine whether they were generalizing insect egg masses as their target odor.

## Imprinting & training

Imprinting refers to early stage odor discrimination training, where the dog is trained that it will be rewarded for sniffing or finding a particular odor. At the beginning of the training, two ports were placed on the floor, one that contained the target odor of dead egg mass on bark, and one that contained an egg mass-free piece of bark as a control. Dogs were imprinted on the target odor by clicking when the dog sniffed the target odor, while trainers ignored any interest in the control bark. On the first day, Pacy received 15 click on sniffs, Grizzly received

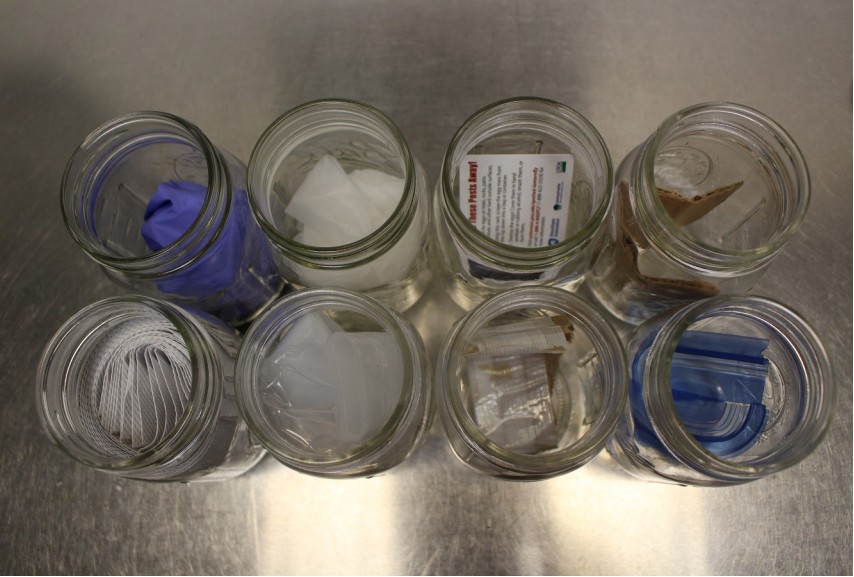

**Fig 6. Picture of the non-bark controls, counterclockwise from top-left: Nitrile gloves, vinyl gloves, plastic scraper, cardboard, paper towels, plastic shipping material, tape, and pieces of plastic storage containers.**

22 click on sniffs, and Toby received 17 click on sniffs. After this first day, no dog was imprinted with the odor on the floor or outside of the wheel. After this short imprinting session, the odors were placed in the scent wheel. If the dogs showed, either by a significant change of behavior or a trained final response, that they recognized the target odor in the wheel, clicking on sniff on the wheel was ceased. During imprinting, one port in the wheel contained the target odor, one port contained an egg mass-free piece of bark, and the other ports contained various non-bark controls.

Initially, one dead egg mass and one egg mass-free bark was used. One more bark control was added once the dogs were greater than 80% successful in a session. We defined success as passing all non-target odors, and showing a trained final response on the target odor, for each trial. Thus, in a 10-trial session, a dog must be successful on 8 trials to move on. Once dogs became proficient on the odor by recognizing this target odor from three other bark controls at 80% success over two consecutive sessions, two dead egg masses were used within each session. This was to ensure that the dogs were not following learning to identify or memorizing one specific odor sample, but following the dead egg mass odor more generally.

### Testing odor–live egg masses

In order to investigate whether the odor of the dead egg masses, and the training the dogs had received, allowed the latter to spontaneously or quickly transfer to live egg masses, we presented the dogs with live egg masses in the same setup as their normal training setup. We presented two egg masses on bark and one egg mass scraped from the bark of two trees of the genus *Morus*, or mulberry (Fig 7A and 7B). These were taken opportunistically from trees located near to the research location. The rest of the scent wheel contained bark and other controls, seven total, as in the training, throughout all trials. As in training, the testing sessions included ten trials total. This initial work on live egg masses was cut short due to the COVID-19 pandemic.

When the lab was able to re-open in the summer of 2020, we resumed training with the three dogs on dead spotted lanternfly egg masses one to two sessions per week, as live ones were no longer available (they were culled during the initial stages of the university lockdown). In October 2020, when live eggs became available, we carried out one test day where the dogs were presented with three novel live spotted lanternfly egg masses, paired with three matched controls of bark from the tree that these masses were taken from, as well as three further new

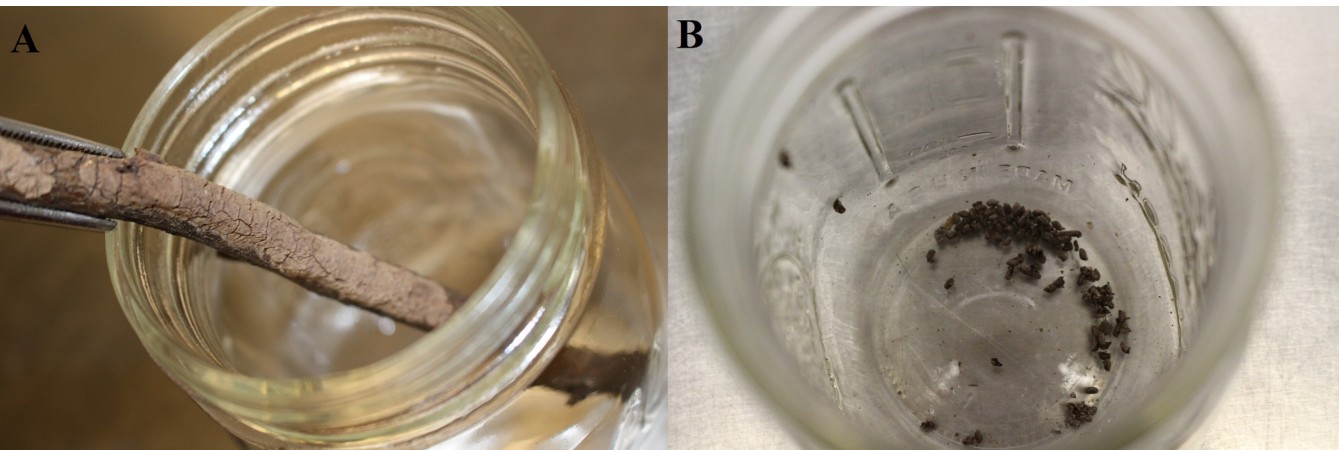

**Fig 7.** (A) Live egg masses on bark and scraped (B).

bark controls, and two non-bark controls (gloves and alcohol). After this test day, dogs were trained on two sessions with novel live spotted lanternfly egg masses, matched controls of bark, and new bark controls, before they were given a second test day. Test days had five total double-blind trials, and where the handler was blinded to the location, as well as presence, of spotted lanternfly egg masses for each trial (some trials were 'blank'). Due to having lower staffing from social distancing measures, training sessions at this point also had fewer sessions than before, typically five trials as well.

## Analysis

Training sessions as well as testing sessions for the dogs were coded by two observers in The Observer XT (Version 14) by Noldus. Training sessions on dead spotted lanternfly prior to the switch to live SLF were coded for each dog (Grizzly = 9 sessions, Pacy = 11 sessions, Toby = 11 sessions), as well as training sessions on live spotted lanternfly before the lab closed for COVID-19 (Grizzly = 4 sessions, Pacy = 4 sessions, Toby = 5 sessions). One week's worth of video from training sessions on live spotted lanternfly before the lab closed for COVID-19 were corrupted and unable to be coded. The two test and two training sessions after the lab returned in the summer were also coded, though one training session for Pacy accidentally not filmed and unable to be coded. Observers were blind to the condition (dead or live spotted lanternfly eggs), what odor was in each port, and coded the videos silenced so they could not hear the 'click' from the trainer. Coders coded for how long each dog spent at each port. Their agreement was high, with an ICC of 0.978.

We calculated the dog's sensitivity (also known as 'True Positive Rate' or TPR, Eq 1) and specificity (also known as 'True Negative Rate' or TNR, Eq 2). These measures are calculated using the number of true positives (the dog correctly showed a final response on the target odor), the number of false negatives (the dog incorrectly passed a non-target odor), and the number of false positives (the dog incorrectly showed a final response on a non-target odor) the dog shows in each session.

Eq 1. Equation for sensitivity.

$$Sensitivity = \frac{\# \ True \ Positives}{(\# \ True \ Positives \ + \# \ False \ Negatives)} \tag{1}$$

**Eq 2.** Equation for specificity.

$$Specificity = \frac{\# \ True \ Negatives}{(\# \ True \ Negatives \ + \# \ False \ Positives)} \tag{2}$$

## Results

All three dogs showed rapid improvement over the first ten sessions. For each session of ten trials for all sessions prior to the COVID-19 lockdown, sensitivity and specificity were calculated. Hesitates on egg masses were used only for informational purposes, and were always noted as incorrect (false negatives). All three dogs had at least one session where both measures were at or above 90%. Averaging all training sessions together, the dogs showed a high level of sensitivity (Grizzly 92.4%, Pacy 79.7%, Toby 94.6%) and specificity (Grizzly 89.3%, Pacy 85.2%, Toby 92.8%). However, there were differences between the three dogs in their levels of consistency. Both Grizzly and Toby were able to maintain relatively consistent performances across training, while Pacy was more variable with her performance (Fig 8A–8C). Pacy had 56 total false negatives compared to nine and 16 for Grizzly and Toby, respectively. The dogs did not show a trained final response on the mantis egg over multiple presentations at any point (Grizzly: 8 passes, Pacy 11 passes, Toby 9 passes).

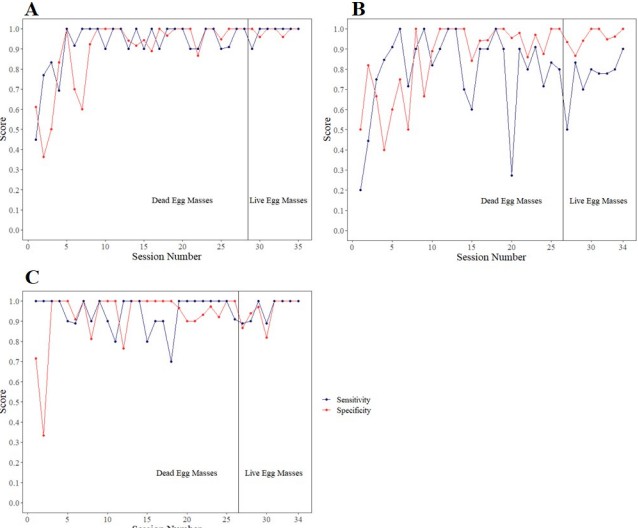

**Fig 8.** Graphs for the sensitivity and specificity for Grizzly (A, top left), Pacy (B, top right), and Toby (C, bottom).

All three dogs spent more time at ports which contained dead spotted lanternfly egg masses than either bark or other controls (Fig 9).

Critically, for the investigation of whether dead egg masses could serve as a training aid for dogs to alert on live egg masses, all three dogs were able to recover their previous performance when the target odor was switched to live spotted lanternfly egg masses within very few sessions (Fig 8A–8C). Each of the three dogs first hesitated on the live egg masses, spending more time sniffing the sample but not giving a full alert at the odor (this was noted as a hesitation). The dogs were not marked at the hesitation, but sent back onto the scent wheel for a second pass. On their second pass, the dogs alerted to the live egg masses, without further imprinting or guidance from their trainers. The dogs alerted to both the live egg masses on bark as well as a live egg mass scraped off bark. Once they were rewarded for this odor, they returned to the same consistency that they showed before on the dead egg masses, over several sessions. Coded data showed that, during training sessions from prior to and post-COVID delays, as with dead spotted lanternfly egg masses, dogs spent more time at ports which contained live spotted lanternfly egg masses than either bark or other controls (Fig 10). This trend was maintained in the double-blind testing sessions as well (Fig 11), where each spotted lanternfly egg

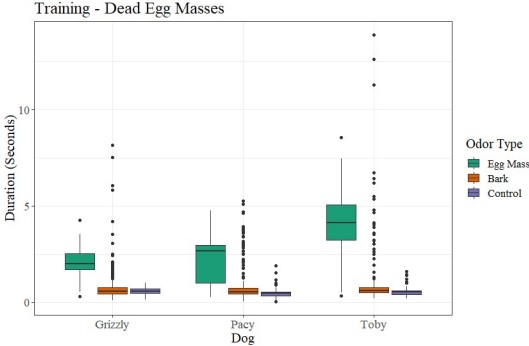

**Fig 9. Duration that each dog spent at each port by what was in each port (either egg masses, bark, or controls) for dead spotted lanternfly egg masses during training.**

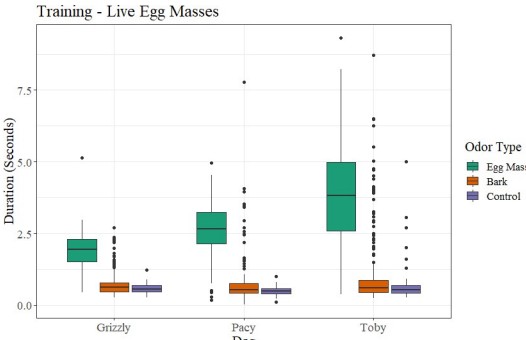

**Fig 10. Duration that each dog spent at each port by what was in each port (either egg masses, bark, or controls) for live spotted lanternfly egg masses during training.**

mass presented was novel and paired with novel bark controls, thus not allowing the dogs to 'follow' any specific target previously rewarded. When tested the second time on fresh spotted lanternfly egg masses, they had not seen fresh egg masses in eight months.

## Discussion

In this study, we have shown that detection dogs trained on dead spotted lanternfly egg masses were able to transfer, though not completely spontaneously, to finding live spotted lanternfly egg masses, both attached to bark as well as scraped from bark without additional odor imprinting. We found that training the dogs to recognize that their target odor was the egg masses, not bark, was the most difficult, though most important, part of the training process for distinguishing the correct odor. Based on the experience with these dogs, dead spotted lanternfly egg masses are an appropriate training aid for detection dogs aiming to find live spotted lanternfly egg masses.

Though the dogs did not spontaneously transfer from the dead spotted lanternfly egg masses to the live ones, the training with the dead egg masses initially allowed for time to train the dogs to distinguish the target odor from the background odor of tree bark without having to maintain live egg masses on-site. This solves two problems in the training process of these dogs: it allows for training outside of the affected areas (where it would be more of a risk to maintain live eggs for training), and it allows for training outside of the time of the year when live egg masses are available. Thus, paired with the quick transfer to live eggs, the dead spotted

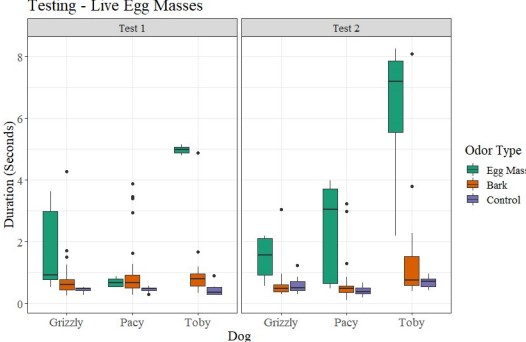

**Fig 11. Duration that each dog spent at each port by what was in each port (either egg masses, bark, or controls) for live spotted lanternfly egg masses during the two double-blind testing sessions, where each spotted lanternfly egg mass presented was novel and paired with novel bark controls.**

lanternfly eggs are a potential training tool for these detection dogs. The COVID-19 pandemic gave us a second chance to test the dogs on spontaneous transfer from dead to live spotted lanternfly egg masses. As we had to take a long hiatus from training, we destroyed all of our live egg masses for the summer, causing us to wait until new egg masses were laid in the fall for further training or testing. The dogs performed well on live spotted lanternfly egg masses, spending over three times longer at ports with egg masses than with bark and five times longer than at ports with non-bark controls, even after not having seen live eggs for eight months and only sporadically training on dead egg masses once we could resume limited social-distanced training. Another option might be to utilize analytical chemistry methods, such as gas chromatography-mass spectrometry, in order to train on scent extract, as was done with another insect species [21]. However, if one is training dogs in an area that is currently under quarantine, and thus there is no risk of spreading the spotted lanternfly to a new area, it may be possible to start the training with live eggs, assuming it is the right time of year.

One dog had difficulty learning the task, and in general could not reach the proficiency and consistency of the other two dogs. This suggests that individual differences between dogs affect the learning process for this odor or task. There is overwhelming evidence in the detection dogs that variability between dogs can affect training as well as eventual success and placement into careers [32–34]. Our training data here suggest that this is true also for potential spotted lanternfly egg detection dogs and may lead handlers to choose different types of dogs for the task. Wallner and Ellis [23] found similarly that some dogs were not appropriate for a similar task with gypsy moth egg masses, and eliminated one dog from their training to focus on the other dogs.

The small sample size, with just three detection dogs going through this training may limit the generalizability to all dogs. However, the different breeds, ages and personalities of the dogs in this study suggest that there may be application to a variety of detection dogs. We were able to document success in all three dogs through training and testing on this odor, and small sample sizes are often used in detection dogs studies, especially during proof of concept [17, 23, 35, 36]. Now that the dead spotted lanternfly egg masses have been shown to be an appropriate training aid, further studies might consider adding more subjects in order to investigate potential differences between dogs' abilities, either through training or in the field work.

The dogs did not learn to just show a final response to insect eggs. After being trained, the dogs were presented with a praying mantis egg mass, and did not show interest in the control odor. Though we were not able to present a number of non-spotted lanternfly egg masses as controls throughout the training process, the fact that the dogs did not show interest in it post-training suggests that they learned the odor of spotted lanternfly egg masses specifically, at least compared to one other species. Future research might investigate whether more closely related insect species' egg masses might be odors for which the dogs should be confirmed to ignore before being utilized in the field.

One limitation to this study is that it did not test the effectiveness of these dogs on their ability to find the egg mass target odor in a real-world scenario, outside of a laboratory setting, after training on this method. Though this was largely outside of the scope of this study, where the aim was to identify a training odor for potential spotted lanternfly detection dogs, we can make some inferences based on our knowledge of detection dogs. We know that dogs have been successful across many types of scent detection [37], including specifically searching cargo [38] and cars [39], which suggests that dogs may be successful in those scenarios when searching for the spotted lanternfly egg masses as well. Moreover, the Pennsylvania Department of Agriculture has now deployed their first spotted lanternfly detection dog to search for egg masses throughout the quarantined counties in their state [40]. Thus, the results presented here may allow this and other future spotted lanternfly detection dogs to train with a safe and effective training aid.

## Supporting information

**S1 File.**
(XLSX)

## Acknowledgments

We would like to thank the staff, interns, and volunteers of the Penn Vet Working Dog Center for their invaluable help in the training and care of our canines and facilities. We would like to thank Amber Bolli for help with coding. We would also like to thank Gregory R. Parra and his team at the USDA for advice, as well as Emily Fricke and her team from the PA Department of Agriculture, for providing samples for training and testing. We would finally like to thank Dr. Julie Urban for her comments on the manuscript.

## Author Contributions

**Conceptualization:** Jennifer L. Essler, Sarah A. Kane, Cynthia M. Otto.

**Data curation:** Jennifer L. Essler, Sarah A. Kane, Amanda Collins, Kaley Ryder.

**Formal analysis:** Jennifer L. Essler, Sarah A. Kane, Amanda Collins.

**Funding acquisition:** Cynthia M. Otto.

**Investigation:** Jennifer L. Essler, Sarah A. Kane, Cynthia M. Otto.

**Methodology:** Jennifer L. Essler, Sarah A. Kane, Amanda Collins, Kaley Ryder, Annemarie DeAngelo, Patricia Kaynaroglu, Cynthia M. Otto.

**Project administration:** Jennifer L. Essler, Cynthia M. Otto.

**Resources:** Cynthia M. Otto.

**Supervision:** Jennifer L. Essler, Annemarie DeAngelo, Patricia Kaynaroglu, Cynthia M. Otto.

**Visualization:** Jennifer L. Essler.

**Writing – original draft:** Jennifer L. Essler.

**Writing – review & editing:** Jennifer L. Essler, Sarah A. Kane, Amanda Collins, Kaley Ryder, Annemarie DeAngelo, Patricia Kaynaroglu, Cynthia M. Otto.

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
