## [Decision Letter · Decision Letter 0]

16 Feb 2021

PONE-D-21-02181

Egg masses as a training aid for spotted lanternfly (Lycorma delicatula) detection dogs

PLOS ONE

Dear Dr. Essler,

Thank you for submitting your manuscript to PLOS ONE. After careful consideration, we feel that it has merit but does not fully meet PLOS ONE’s publication criteria as it currently stands. Therefore, we invite you to submit a revised version of the manuscript that addresses the points raised during the review process.

Throughout the manuscript, the authors focused more on the dogs used to detect egg masses of the spotted lanternfly, but they forgot in some parts to highlight important aspects supported by relevant references (biology, behavioral ecology, host plants,...) linked to the invasive insect, target of the present study. This might be obvious (but not sufficient to reach article acceptance) considering the authors are not entomologists. Therefore, it is mandatory to make major revisions regarding this issue through using correct entomological terms and aspects, adding all missing info, and deleting redundant and confusing statements. Furthermore, the potential field effectiveness (and possible natural constraints) of the detection method being investigated here should be discussed by the authors. All reviewers' comments are stated below.

We look forward to receiving your revised manuscript.

Kind regards,

Ramzi Mansour

Academic Editor

PLOS ONE

Journal Requirements:

2. Please include captions for your Supporting Information files at the end of your manuscript, and update any in-text citations to match accordingly. Please see our Supporting Information guidelines for more information: http://journals.plos.org/plosone/s/supporting-information

Reviewers' comments:

Reviewer's Responses to Questions

**Comments to the Author**

1. Is the manuscript technically sound, and do the data support the conclusions?

Reviewer #1: Yes

Reviewer #2: Partly

2. Has the statistical analysis been performed appropriately and rigorously? 

Reviewer #1: N/A

Reviewer #2: N/A

3. Have the authors made all data underlying the findings in their manuscript fully available?

Reviewer #1: Yes

Reviewer #2: Yes

4. Is the manuscript presented in an intelligible fashion and written in standard English?

Reviewer #1: Yes

Reviewer #2: Yes

5. Review Comments to the Author

Reviewer #1: 02/04/2021

Spotted lanternfly (SLF) is an exotic pest of various agricultural and forest crops in North America. The current research deals with the potential of dogs in the detection of the egg masses. It’s an exciting small study that could offer a new tool for the fast and reliable survey of this pest. It should be of interest to this journal. However, the entire manuscript suffered from the lack of discipline in writing. Mistakes in the description of SLF biology will also need to be corrected before it can be considered for publication.

Major concerns:

1. Write scientifically by using clear and concise sentences. Stay focus on the subject matter for each paragraph. Exclude unrelated information. Don’t mix up information from different subjects, e.g., information belongs to Material and Methods should not be included in Results, etc. I have made some suggestions below, but more thorough editing is needed to improve the manuscript.

2. Understanding none of the authors were trained in entomology, critical review of the manuscript from a qualified entomologist is strongly recommended to ensure proper use of nomenclature and correct description of other facts about this pest.

3. The introduction portion of SLF biology (ln 55-72) need a complete overhaul, with the first paragraph focus on hosts and damages, and the second paragraph on life cycle and egg stage.

4. SLF egg masses are mostly found on tree trunks and upper branches, with a small proportion on stones and other nonliving substrates on the ground. I do not see canines being actively used to detect egg masses in real field situations. Maybe more useful in cargo and commodity inspections as part of the quarantine enforcement?

5. How do you match the controls? Ideally, you should match target odor with control odor from the same tree species for both dead and live egg masses. Was that how you set up the trials? More explanation is needed.

Specific comments:

1. No need to credit photos taken by one of the co-authors.

2. ln 1. Should be “Egg mass” not “Egg Masses”.

3. ln 33-34. Awkward writing. Rephrase please.

4. ln 56. “Vietnam”. Not true anymore. Delete.

5. ln 57. “eating the phloem”. Not true. SLF is a sap feeder. Also, there is no need to include the definition of phloem.

6. ln 58-60. Awkward writing. Rephrase please.

7. ln 65. Confusing.

8. ln 84-86. This does not belong here. Move to the last paragraph n introduction as part of the study objectives.

9. ln 109-111. This should be included in the “Ethics Statement” with a few more words on the use of the canines.

10. ln 196-197. Citation needed.

11. ln 284-286. Delete. Already described in M&M.

12. ln 310-314. These equations belong to M&M, not here.

13. ln 359. “immediately spontaneously”. Awkward wording. Re-word.

14. ln 362-365. Good point.

15. ln 373-375. Would be interesting to see what’s responsible for the scent extract.

Reviewer #2: In this study, the authors present detection dogs as a method to find Lycorma delicatula egg mass. Throughout the manuscript, the methodology of training detection dog is presented and analyzed well. Nevertheless, the manuscript contains two major issues to be resolved before the article is to be published: an in-depth consideration of the target organism itself and detection efficacy of the method in the field.

First, although the paper focuses on the training method of detection dogs, it does not have deep enough consideration on the target organism itself. Although the paper describes general phenology of the insect, we were unable to find information on the oviposition behavior or characteristics of egg mass itself, which is the target of detection. For example, what is the major oviposition host plant of L. delicatula, or at which height or substrate are the egg masses often be found? Without the considering the biology of L. delicatula and its egg mass, we may not be able to apply the method in the field.

Second, there is no field testing of the method presented in the study. As L. delicatula lays eggs on a variety of substrates including metal fence or host plants, the efficacy of the method may vary on the environment. Furthermore, recent study suggests that L. delicatula eggs can be found on tree-of-heaven or black walnut trees from up to 18 m above the ground (Liu and Hartlieb 2020). Especially, most favored egg heights were 8-10 m for tree-of-heaven and 4-6 m for black walnut. In this case, can dogs detect eggs at this height, or even if it does, is there a way to confirm its presence? The paper at least should address a more in-depth concerns on lack of efficacy experiment.

The following contains comments on the manuscript:

Manuscript

1) L65 – 72: Although the paragraph describes oviposition of L. delicatula, it lacks a more in-depth information on egg mass, which is the target of detection. For example, what is the most preferred host plant/ substrate? How big is an egg mass? Is there just one egg mass per tree or can more than one egg mass be found? What are the characteristics of the egg mass substrate (height/ surface texture etc.)

2) L407 – 412: It describes that further research is required to test the efficacy of the detection method. Nevertheless, we would appreciate if this can be described in more details. For example, how confident are you that the method will work perfectly outdoor as well? Where would you find L. delicatula egg masses and what potential factors may limit the application of the methods? Also, if you find them, what is to be done for control of the egg masses? Limitation of the manuscript needs to be discussed in more detail.

3) Few editorial corrections are to be made in references – scientific names of insects need to be italicized

6. PLOS authors have the option to publish the peer review history of their article (what does this mean?). If published, this will include your full peer review and any attached files.

Reviewer #1: No

Reviewer #2: No

---

## [Author Response · Author response to Decision Letter 0]

22 Mar 2021

The following is included with proper formatting in the "Response to Reviewers" file.

Reviewer #1: 02/04/2021

Spotted lanternfly (SLF) is an exotic pest of various agricultural and forest crops in North America. The current research deals with the potential of dogs in the detection of the egg masses. It’s an exciting small study that could offer a new tool for the fast and reliable survey of this pest. It should be of interest to this journal. However, the entire manuscript suffered from the lack of discipline in writing. Mistakes in the description of SLF biology will also need to be corrected before it can be considered for publication.

Major concerns:

1. Write scientifically by using clear and concise sentences. Stay focus on the subject matter for each paragraph. Exclude unrelated information. Don’t mix up information from different subjects, e.g., information belongs to Material and Methods should not be included in Results, etc. I have made some suggestions below, but more thorough editing is needed to improve the manuscript.

Thank you, the manuscript has been edited for clarity.

2. Understanding none of the authors were trained in entomology, critical review of the manuscript from a qualified entomologist is strongly recommended to ensure proper use of nomenclature and correct description of other facts about this pest.

We have spoken with Dr. Julie Urban, who works with SLF extensively, and have made comments based off of her reading of the manuscript as well. She is also acknowledged the acknowledgments.

3. The introduction portion of SLF biology (ln 55-72) need a complete overhaul, with the first paragraph focus on hosts and damages, and the second paragraph on life cycle and egg stage.

4. SLF egg masses are mostly found on tree trunks and upper branches, with a small proportion on stones and other nonliving substrates on the ground. I do not see canines being actively used to detect egg masses in real field situations. Maybe more useful in cargo and commodity inspections as part of the quarantine enforcement?

It is true that since the drafting of this MS and the introduction of an SLF detection canine by PA Dept of Ag, implementation of this work has been utilized in cargo and commodity inspections as well as nurseries. We have modified the manuscript to better capture that.

5. How do you match the controls? Ideally, you should match target odor with control odor from the same tree species for both dead and live egg masses. Was that how you set up the trials? More explanation is needed.

In general, with the training sets, we did not always have the knowledge of the exact tree that a mass was from. We did include controls of the same species of tree. In particular with the double-blind testing, the controls were from the same exact tree as the egg mass. This is now more clearly expressed in the manuscript.

Specific comments:

1. No need to credit photos taken by one of the co-authors.

Photo credit has been removed.

2. ln 1. Should be “Egg mass” not “Egg Masses”.

Reworded title.

3. ln 33-34. Awkward writing. Rephrase please.

Modified writing.

4. ln 56. “Vietnam”. Not true anymore. Delete.

Thank you – removed.

5. ln 57. “eating the phloem”. Not true. SLF is a sap feeder. Also, there is no need to include the definition of phloem.

Given our discussion with Dr. Urban we have modified this section and the information on the SLF generally.

6. ln 58-60. Awkward writing. Rephrase please.

Modified.

7. ln 65. Confusing.

Modified.

8. ln 84-86. This does not belong here. Move to the last paragraph n introduction as part of the study objectives.

This sentence is about the Wallner and Ellis study from the prior sentence and has been modified to be clearer.

9. ln 109-111. This should be included in the “Ethics Statement” with a few more words on the use of the canines.

We have added an “Ethics Statement” section within the Methods section of the manuscript.

10. ln 196-197. Citation needed.

We have updated this sentence and added a citation.

11. ln 284-286. Delete. Already described in M&M.

Deleted.

12. ln 310-314. These equations belong to M&M, not here.

Moved.

13. ln 359. “immediately spontaneously”. Awkward wording. Re-word.

Removed “immediately”.

14. ln 362-365. Good point.

15. ln 373-375. Would be interesting to see what’s responsible for the scent extract.

Reviewer #2: In this study, the authors present detection dogs as a method to find Lycorma delicatula egg mass. Throughout the manuscript, the methodology of training detection dog is presented and analyzed well. Nevertheless, the manuscript contains two major issues to be resolved before the article is to be published: an in-depth consideration of the target organism itself and detection efficacy of the method in the field.

First, although the paper focuses on the training method of detection dogs, it does not have deep enough consideration on the target organism itself. Although the paper describes general phenology of the insect, we were unable to find information on the oviposition behavior or characteristics of egg mass itself, which is the target of detection. For example, what is the major oviposition host plant of L. delicatula, or at which height or substrate are the egg masses often be found? Without the considering the biology of L. delicatula and its egg mass, we may not be able to apply the method in the field.

We agree with this comment, and have reframed the introduction to describe search scenarios that dogs are more likely to be utilized for, including nurseries, cargo, and automobiles, rather than trees specifically where, it is true, many of the egg masses are far too high up for the dogs to (likely) be especially effective.

Second, there is no field testing of the method presented in the study. As L. delicatula lays eggs on a variety of substrates including metal fence or host plants, the efficacy of the method may vary on the environment. Furthermore, recent study suggests that L. delicatula eggs can be found on tree-of-heaven or black walnut trees from up to 18 m above the ground (Liu and Hartlieb 2020). Especially, most favored egg heights were 8-10 m for tree-of-heaven and 4-6 m for black walnut. In this case, can dogs detect eggs at this height, or even if it does, is there a way to confirm its presence? The paper at least should address a more in-depth concerns on lack of efficacy experiment.

We agree with this and like above, it is unlikely that dogs used for this will be searching trees, but rather cargo, rail and cars, among other lower items. We have included in the introduction more specifics about the likely implementation of these dogs, once trained.

The following contains comments on the manuscript:

Manuscript

1) L65 – 72: Although the paragraph describes oviposition of L. delicatula, it lacks a more in-depth information on egg mass, which is the target of detection. For example, what is the most preferred host plant/ substrate? How big is an egg mass? Is there just one egg mass per tree or can more than one egg mass be found? What are the characteristics of the egg mass substrate (height/ surface texture etc.)

We have modified and added to the introduction significantly, including these suggestions. 

2) L407 – 412: It describes that further research is required to test the efficacy of the detection method. Nevertheless, we would appreciate if this can be described in more details. For example, how confident are you that the method will work perfectly outdoor as well? Where would you find L. delicatula egg masses and what potential factors may limit the application of the methods? Also, if you find them, what is to be done for control of the egg masses? Limitation of the manuscript needs to be discussed in more detail.

We have modified the discussion to include more appropriately the likely use of detection dogs in this arena, and how the data presented here may help future spotted lanternfly detection dogs.

3) Few editorial corrections are to be made in references – scientific names of insects need to be italicized

References have been modified.

---

## [Decision Letter · Decision Letter 1]

6 Apr 2021

PONE-D-21-02181R1

Egg masses as training aids for spotted lanternfly (Lycorma delicatula) detection dogs

PLOS ONE

Dear Dr. Essler,

Thank you for submitting your manuscript to PLOS ONE. After careful consideration, we feel that it has merit but does not fully meet PLOS ONE’s publication criteria as it currently stands. Therefore, we invite you to submit a revised version of the manuscript that addresses the points raised during the review process.

We look forward to receiving your revised manuscript.

Kind regards,

Ramzi Mansour

Academic Editor

PLOS ONE

Journal Requirements:

Reviewers' comments:

Reviewer's Responses to Questions

**Comments to the Author**

1. If the authors have adequately addressed your comments raised in a previous round of review and you feel that this manuscript is now acceptable for publication, you may indicate that here to bypass the “Comments to the Author” section, enter your conflict of interest statement in the “Confidential to Editor” section, and submit your "Accept" recommendation.

Reviewer #1: All comments have been addressed

Reviewer #2: All comments have been addressed

2. Is the manuscript technically sound, and do the data support the conclusions?

Reviewer #1: Yes

Reviewer #2: Yes

3. Has the statistical analysis been performed appropriately and rigorously? 

Reviewer #1: Yes

Reviewer #2: Yes

4. Have the authors made all data underlying the findings in their manuscript fully available?

Reviewer #1: Yes

Reviewer #2: Yes

5. Is the manuscript presented in an intelligible fashion and written in standard English?

Reviewer #1: Yes

Reviewer #2: Yes

6. Review Comments to the Author

Reviewer #1: 03/26/2021

All previous concerns have been properly addressed by the authors. Therefore, I recommend its publication by the journal with a few minor changes below:

1. At the first citation of a genus/species, authority, order, and family should be given in parentheses.

2. Be consistent when using the insect or plant names, e.g., don’t not use “The spotted lanternfly” in one sentence, and use “SLF” in another.

3. ln 33. Replace “The spotted lanternfly” with “It” since it was used in the previous sentence.

4. ln 192-193. I believe the correct common name for Prunus serrullata is Japanese flowering cherry.

5. ln 271, 274, 277, etc. acronym “SLF” was not defined and therefore should not be used.

Reviewer #2: Overall, the authors revised the manuscript according to the comments made by the reviewers. Once the following minor points are addressed by author, the manuscript is ready to be published.

First, this is a minor recommendation that may improve the manuscript. Because the paper focuses on detection of L. delicatula egg mass on artificial substrates including cargo, rail, and automobiles it may strengthen the importance of screening for these substrates by highlighting the potential spread of L. delicatula via these transportation means. This can be done by 1) citing different invasive insects that rapidly expanded its geographical range by hijacking trains and automobiles, 2) citing previous studies that observed potential L. delicatula dispersal via train (Flight Dispersal Capabilities of Female Spotted Lanternflies (Lycorma delicatula) Related to Size and Mating Status, Wolfin et al. 2019).

Second, the authors must describe the order and family of an insect when first mentioning its scientific name for all the insects. E.g. Lycorma delicatula (Hemipera: Fulgoridae). Also, if scientific name of an organism has already been mentioned, abbreviate its generic name in second time. E.g. Ailanthus altissima -> A. altissima.

Finally, the following comments address some minor editorial changes to be made and confusing sentences that need to be rephrased.

1. Ln 55: missing a period

2. Ln 68: space

3. Ln 72 – 75: confusing. Rephrase.

4. Ln 82 – 84: rephrase

5. Ln 101 – 104: add references

6. Ln 107 – 109: rephrase

7. Ln 360 – 362: rephrase

7. PLOS authors have the option to publish the peer review history of their article (what does this mean?). If published, this will include your full peer review and any attached files.

Reviewer #1: No

Reviewer #2: No

---

## [Author Response · Author response to Decision Letter 1]

12 Apr 2021

Reviewer #1

All previous concerns have been properly addressed by the authors. Therefore, I recommend its publication by the journal with a few minor changes below:

1. At the first citation of a genus/species, authority, order, and family should be given in parentheses.

We have added (Order: Family) for each insect and tree species noted in the manuscript, on their first citation.

2. Be consistent when using the insect or plant names, e.g., don’t not use “The spotted lanternfly” in one sentence, and use “SLF” in another.

We have replaced “SLF” in the manuscript with spotted lanternfly.

3. ln 33. Replace “The spotted lanternfly” with “It” since it was used in the previous sentence.

Replaced.

4. ln 192-193. I believe the correct common name for Prunus serrullata is Japanese flowering cherry.

Thank you – replaced.

5. ln 271, 274, 277, etc. acronym “SLF” was not defined and therefore should not be used.

The acronym “SLF” was removed throughout the manuscript and replaced.

Reviewer # 2

Overall, the authors revised the manuscript according to the comments made by the reviewers. Once the following minor points are addressed by author, the manuscript is ready to be published.

First, this is a minor recommendation that may improve the manuscript. Because the paper focuses on detection of L. delicatula egg mass on artificial substrates including cargo, rail, and automobiles it may strengthen the importance of screening for these substrates by highlighting the potential spread of L. delicatula via these transportation means. This can be done by 1) citing different invasive insects that rapidly expanded its geographical range by hijacking trains and automobiles, 2) citing previous studies that observed potential L. delicatula dispersal via train (Flight Dispersal Capabilities of Female Spotted Lanternflies (Lycorma delicatula) Related to Size and Mating Status, Wolfin et al. 2019). 

Specific highlight of this species (and invasive insects in general) using railways as a means of transport has been added to the introduction, thank you.

Second, the authors must describe the order and family of an insect when first mentioning its scientific name for all the insects. E.g. Lycorma delicatula (Hemipera: Fulgoridae). Also, if scientific name of an organism has already been mentioned, abbreviate its generic name in second time. E.g. Ailanthus altissima -> A. altissima.

Thank you – the order and family for the spotted lanternfly were added, and scientific names were abbreviated where necessary.

Finally, the following comments address some minor editorial changes to be made and confusing sentences that need to be rephrased.

1. Ln 55: missing a period

2. Ln 68: space

3. Ln 72 – 75: confusing. Rephrase.

4. Ln 82 – 84: rephrase

5. Ln 101 – 104: add references

6. Ln 107 – 109: rephrase

7. Ln 360 – 362: rephrase

Thank you – all of the above have been modified.

---

## [Editor Report · Decision Letter 2]

19 Apr 2021

Egg masses as training aids for spotted lanternfly (Lycorma delicatula) detection dogs

PONE-D-21-02181R2

Dear Dr. Essler,

We’re pleased to inform you that your manuscript has been judged scientifically suitable for publication (please see ADDITIONAL EDITOR COMMENTS below) and will be formally accepted for publication once it meets all outstanding technical requirements.

Kind regards,

Ramzi Mansour

Academic Editor

PLOS ONE

Additional Editor Comments:

The following minor revisions should be made by the authors on the PROOFS of their accepted article, before its publication:

L2 (Title): delete the brackets (replace "(Lycorma delicatula)" with "Lycorma delicatula"

L33 (Abstract):  to avoid repetition, replace "The spotted lanternfly feeds"  with  "This insect feeds"

L33-34:  replace "severe damage to vineyards and leaves a sooty mold that"   with  "severe damage in vineyards such as the occurrence of sooty mold fungus that"

L35: replace "It eggs that"  with  "It lays eggs that"

L39: change "this overwinter period"  to  "the overwintering period"

L53: add the authorship "(White)" to the species and delete the brackets as indicated here: The spotted lanternfly Lycorma delicatula (White) (Hemiptera: Fulgoridae)

L 56: change to  "and further damage is induced by"

L57: delete the “ ” (this should be written: the honeydew without “ ” )

L69: change to "their most preferred host plant [6]"

L71: delete the " . "  after  "ground"

L73-74: replace "and those in the field are taking" with "and technical professionals in the field are taking"

L78:  replace "could make a significant impact on this problem" with "could be a promising pest management approach, reducing its serious damage on host plants"

L84: change to "during the overwintering period"

L93: add the authorship "L." to the species: Lymantria dispar L.

L94: In that study, dogs

L96: of citrus huanglongbing (citrus greening disease) long before

L207: plant from the genus

L212: write the full species as this is a caption: "Figure 5. Picture of one............from Ailanthus altissima"

L216-217: Barks from other plant species were introduced at

L227: add the authorship "(L.)" to the species: Mantis religiosa (L.)

L261: allowed the latter to spontaneously or

L264: genus Morus

L269: there are no letters "A" and "B" on this Figure 7, please add both letters, and change the caption to "Figure 7A-B. (A) Live egg masses on bark and (B) scraped.

L311: All three dogs showed rapid

L325-326: add the three letters "A" , "B" and "C" on the Figure 8 (no letters can be seen on this figure)

L407 : delete "generally"

L419: delete "generally"
---

## [Editor Report · Acceptance letter]

22 Apr 2021

PONE-D-21-02181R2 

Egg masses as training aids for spotted lanternfly *Lycorma delicatula* detection dogs 

Dear Dr. Essler:

I'm pleased to inform you that your manuscript has been deemed suitable for publication in PLOS ONE. Congratulations! Your manuscript is now with our production department. 

Kind regards, 

on behalf of

Dr. Ramzi Mansour 

Academic Editor

PLOS ONE